# A Simple Cache Model for Image Recognition

**Emin Orhan**
`aeminorhan@gmail.com`
Baylor College of Medicine & New York University

## Abstract

Training large-scale image recognition models is computationally expensive. This raises the question of whether there might be simple ways to improve the test performance of an already trained model without having to re-train or fine-tune it with new data. Here, we show that, surprisingly, this is indeed possible. The key observation we make is that the layers of a deep network close to the output layer contain independent, easily extractable class-relevant information that is not contained in the output layer itself. We propose to extract this extra class-relevant information using a simple key-value cache memory to improve the classification performance of the model at test time. Our cache memory is directly inspired by a similar cache model previously proposed for language modeling (Grave et al., 2017). This cache component does not require any training or fine-tuning; it can be applied to any pre-trained model and, by properly setting only two hyper-parameters, leads to significant improvements in its classification performance. Improvements are observed across several architectures and datasets. In the cache component, using features extracted from layers close to the output (but not from the output layer itself) as keys leads to the largest improvements. Concatenating features from multiple layers to form keys can further improve performance over using single-layer features as keys. The cache component also has a regularizing effect, a simple consequence of which is that it substantially increases the robustness of models against adversarial attacks.

## 1  Introduction

Deep neural networks are currently the state of the art models in a wide range of image recognition problems. In the standard supervised learning setting, training these models typically requires a large number of labeled examples. This causes at least two potential problems. First, the large model and training set sizes make it computationally expensive to train these models. Thus, it would be desirable to ensure that the model's performance is as good as it can be, given a particular budget of training data and training time. Secondly, these models might have difficulties in cases where correct classification depends on the detection of rare but distinctive features in an image that do not occur frequently enough in the training data. In this paper, we propose a method that addresses both of these problems.

Our key observation is that the layers of a deep neural network close to the output layer contain independent, easily extractable class-relevant information that is not already contained in the output layer itself. We propose to extract this extra class-relevant information with a simple key-value cache memory that is directly inspired by Grave et al. [3], where a similar cache model was introduced in the context of language modeling.

Our model addresses the two problems described above. First, by properly setting only two hyper-parameters, we show that a pre-trained model's performance at test time can

be improved significantly without having to re-train or even fine-tune it with new data. Secondly, storing rare but potentially distinctive features in a cache memory enables our model to successfully retrieve the correct class labels even when those features do not appear very frequently in the training data.

Finally, we show that the cache memory also has a regularizing effect on the model. It achieves this effect by increasing the size of the input region over which the model behaves similarly to the way it behaves near training data and prior work has shown that trained neural networks behave more regularly near training data than elsewhere in the input space [11]. A useful consequence of this effect is the substantially improved robustness of cache models against adversarial attacks.

## 2  Results

Our cache component is conceptually very similar to the cache component proposed by Grave et al. [3] in the context of language modeling. Following [3], we define a cache component by a pair of key and value matrices $(\mu, \upsilon)$. Here, $\mu$ is a $d \times K$ matrix of keys where $K$ is the number of items stored in the cache and $d$ is the dimensionality of the key vectors, $\upsilon$ is a $C \times K$ matrix of values where $C$ is the number of classes. We use $\mu_k$ and $\upsilon_k$ to denote the $k$-th column of $\mu$ and $\upsilon$, respectively.

To build a key matrix $\mu$, we pass the training data (or a subset of it) through an already trained network. The key vector $\mu_k$ for a particular item $\mathbf{x}_k$ in the training set is obtained by taking the activities of one or more layers of the network when $\mathbf{x}_k$ is input to the network, vectorizing those activities, and concatenating them if more than one layer is used (Figure 1). We then normalize the resulting key vector to have unit norm. The value vector $\upsilon_k$ is simply the one-hot encoding of the class label for $\mathbf{x}_k$.

Given a test item $\mathbf{x}$ with label $\mathbf{y}$ (considered as a one-hot vector), its similarity with the stored items in the cache is computed as follows:

$$\sigma_k(\mathbf{x}) \propto \exp(\theta \phi(\mathbf{x})^{\top} \mu_k) \tag{1}$$

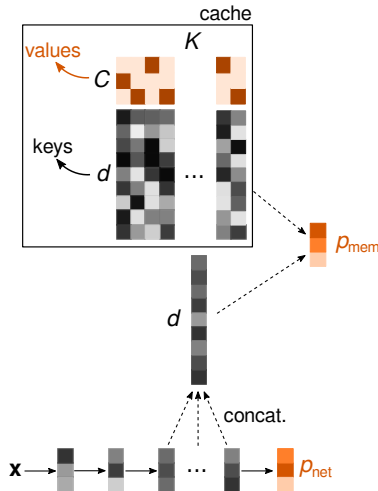

We use $\phi(\mathbf{x})$ to denote the representation of $\mathbf{x}$ in the layers used in the cache component. A distribution over labels is obtained by taking a weighted average of the values stored in the cache:

$$p_{\mathrm{mem}}(\mathbf{y}|\mathbf{x}) = \frac{\sum_{k=1}^{K} \upsilon_k \sigma_k(\mathbf{x})}{\sum_{k=1}^{K} \sigma_k(\mathbf{x})} \tag{2}$$

The hyper-parameter $\theta$ in Equation 1 controls the sharpness of this distribution, with larger $\theta$ values producing sharper distributions. This distribution is then combined with the usual forward model implemented in the final softmax layer of the trained network:

$$p(\mathbf{y}|\mathbf{x}) = (1 - \lambda)p_{\mathrm{net}}(\mathbf{y}|\mathbf{x}) + \lambda p_{\mathrm{mem}}(\mathbf{y}|\mathbf{x}) \tag{3}$$

Figure 1: Schematic diagram of the cache model.

Here, $\lambda$ controls the overall weight of the cache component in the model. The model thus has only two hyper-parameters, i.e. $\theta$ and $\lambda$, which we optimize through a simple grid-search procedure on held-out validation data (we search over the ranges $10 \le \theta \le 90$ and $0.1 \le \lambda \le 0.9$).

**Extracting additional class-relevant information from a pre-trained model**

As our baseline models, we used deep ResNet [5] and DenseNet [7] models trained on the CIFAR-10, CIFAR-100, and ImageNet (ILSVRC2012) datasets. Standard data augmentation was used to double the training set size for the CIFAR-10 and CIFAR-100 datasets. For CIFAR-10 and CIFAR-100, we used the training set to train the models, the validation set to optimize the hyper-parameters of the cache models and finally reported the error rates

| Model | Params | C-10+ | C-100+ | ImageNet |
|---|---|---|---|---|
| ResNet20 ($\lambda = 0$) | 0.27M | 8.33 | 32.66 | – |
| ResNet20-Cache3 | 0.27M | **7.58** | **29.18** | – |
| ResNet20-Cache3-CacheOnly ($\lambda = 1$) | 0.27M | 11.87 | 39.18 | – |
| ResNet32 ($\lambda = 0$) | 0.46M | 7.74 | 32.94 | – |
| ResNet32-Cache3 | 0.46M | **7.01** | **29.36** | – |
| ResNet32-Cache3-CacheOnly ($\lambda = 1$) | 0.46M | 11.13 | 38.40 | – |
| ResNet56 ($\lambda = 0$) | 0.85M | 8.74 | 31.11 | – |
| ResNet56-Cache3 | 0.85M | **8.11** | **27.99** | – |
| ResNet56-Cache3-CacheOnly ($\lambda = 1$) | 0.85M | 13.05 | 34.06 | – |
| DenseNet40 ($\lambda = 0$) | 1M | 5.75 | 27.08 | – |
| DenseNet40-Cache2 | 1M | **5.44** | **25.25** | – |
| DenseNet40-Cache2-CacheOnly ($\lambda = 1$) | 1M | 8.68 | 37.64 | – |
| DenseNet100 ($\lambda = 0$) | 7M | 5.08 | 22.62 | – |
| DenseNet100-Cache1 | 7M | **4.92** | **21.95** | – |
| DenseNet100-Cache1-CacheOnly ($\lambda = 1$) | 7M | 7.44 | 32.74 | – |
| ResNet50 ($\lambda = 0$) | 25.6M | – | – | 30.98 |
| ResNet50-Cache1 | 25.6M | – | – | **30.42** |
| ResNet50-Cache1-CacheOnly ($\lambda = 1$) | 25.6M | – | – | 41.24 |

Table 1: Error rates of different models on CIFAR-10, CIFAR-100, and ImageNet datasets (+ indicates the standard data augmentation for the CIFAR datasets). In the cache models, the number next to "Cache" represents the number of layers used for constructing the key vectors: e.g. "Cache3" means 3 different layers were concatenated in creating the key vectors. The results for ImageNet are top-1 error rates. We did not run separate layer searches for the cache-only models ("CacheOnly"); these models used the same layers as the corresponding linear-combination cache models (the hyper-parameter $\theta$, however, was optimized separately for these models).

on the test set. For the ImageNet dataset, we took a pre-trained ResNet50 model, split the validation set into two, used the first half to optimize the hyper-parameters of the cache models and reported the error rates on the second half.

We compared the performance of the baseline models with the performance of two types of cache model. The first one is the model described in Equation 3 above, where the predictions of the cache component and the network are linearly combined. The second type of model is a cache-only model where we just use the predictions of the cache component, i.e. we set $\lambda = 1$ in Equation 3. This cache-only model thus has a single hyper-parameter, $\theta$, that has to be optimized (Equation 1).

For the CIFAR-10 and CIFAR-100 datasets, we used all items in the (augmented) training set to generate the keys stored in the cache (90K items in total). For the ImageNet dataset, using the entire training set was not computationally feasible, hence we used a random subset of 275K items (275 items per class) from the training set to generate the keys. This corresponds to approximately 22% of the training set. For the cache models where activations in only a single layer were used as keys, the optimal layer was chosen with cross-validation on held-out data by sweeping through all layers in the network. In cases where activations in multiple layers were concatenated to generate the keys, sweeping through all combinations of layers was not feasible, hence we used manual exploration to search over the space of relevant combinations of layers. Whenever possible, we tried combinations of up to 3 layers in the network and report the test performance of the model that yielded the best accuracy on the validation data.

Table 1 shows the error rates of the models with or without a cache component on the three datasets.[1] In all cases, the cache model that linearly combines the predictions of the network and the cache component has the highest accuracy. The fact that cache models perform

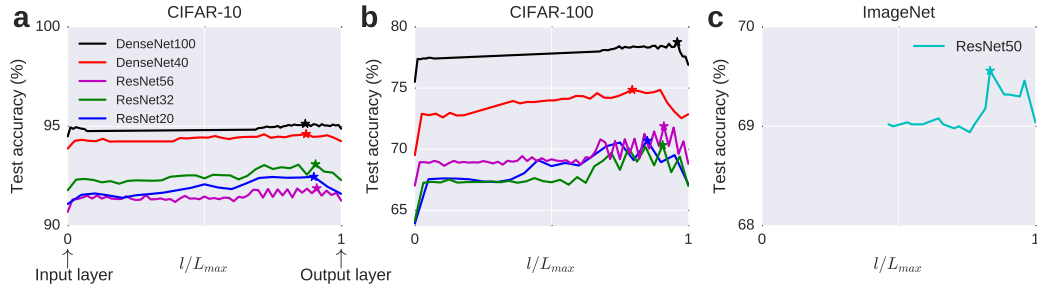

Figure 2: When used as key vectors in the cache component, layers close to the output, but not the output layer itself, lead to the largest improvements in accuracy. Here, test accuracy is plotted against the normalized layer index (with 0 indicating the input layer, i.e. the image, and 1 indicating the output layer) for different models. The best value for each model is indicated with a star symbol ($\star$). Results are shown for the (a) CIFAR-10, (b) CIFAR-100, and (c) ImageNet datasets. In (c), only the higher layers were low-dimensional enough to be used as key vectors, hence results are shown only for mid-network and upper layers. In these experiments, we set the hyper-parameters to the middle of the ranges over which they were allowed to vary.

significantly better than the baseline models, which only use the output layer of the network to make predictions, suggests that layers other than the output layer contain independent class-relevant information and that this extra information can be easily read-out using a simple continuous key-value memory (Equations 1-3).

To find out which layers contained this extra class-relevant information, we tried using different layers of the network, all the way from the input layer, i.e. the image itself, to the output layer, as key vectors in the cache component. This analysis showed that layers close to the output contained most of the extra class-relevant information (Figure 2). Using the output layer itself in the cache component resulted in test performance comparable to that of the baseline model. Using the input layer, on the other hand, generally resulted in worse test performance than the baseline (Figure 2a-b). This is presumably because when the input layer, or other low-level features in general, are used as key vectors, the model becomes susceptible to surface similarities that are not relevant for the classification task. Higher-level features, on the other hand, are less likely to be affected by such superficial similarities.

The cache-only models performed worse than the baseline models (Table 1, "CacheOnly"). However, as we show below, these models turned out to be more robust to input perturbations than both the baseline models and the linear-combination cache models. Hence, they may be preferred in cases where it is desirable to trade off a certain amount of accuracy for improved robustness.

To investigate the effect of cache size on the models' performance, we varied the cache size from 0% of the training data (i.e. no cache) to 100% of the training data (i.e. using the entire training data for the cache). Importantly, we optimized the hyper-parameters of the cache models separately for each cache size. Representative results are shown in Figure 3 for ResNet32 models trained on CIFAR-10 and CIFAR-100. We observed significant improvements in test accuracy over the baseline model even with small cache sizes and the performance of the cache models increased steadily with the cache size.

**Cache component improves the robustness of image recognition models**

One possible way to view the cache component is as implementing a prior over images that favors images similar to those stored in the cache memory, i.e. training images, where the relevant notion of similarity is based on some high-level features of the images. The cache component makes a model's response to new images more similar to its response to the training images. It has been shown that trained deep neural networks behave more regularly

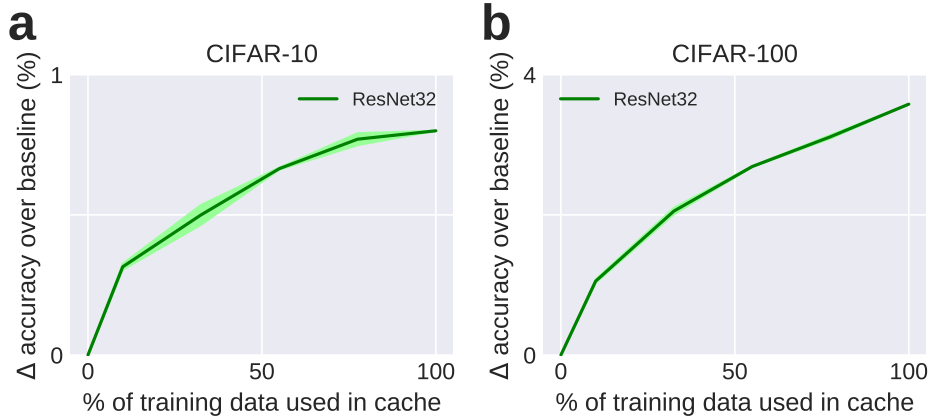

Figure 3: The effect of cache size on the test accuracy of a representative model trained on CIFAR-10 (a) and CIFAR-100 (b). Shaded regions represent ±1 s.e.m. over 2 independent runs. Note that 0% on the $x$-axis corresponds to the baseline model with no cache component.

in the neighborhood of training data than elsewhere in the input space [11]: they have smaller Jacobian norms and a smaller number of "linear regions" near the training data, which enables decreased sensitivity to perturbations and hence better generalization in the neighborhood of training points in the input space. This suggests that by effectively imposing a prior over the input space that favors the training data, a cache component may extend the range over which the model behaves regularly and hence improve its generalization behavior outside the local neighborhood of the training data. Potentially, this also includes improved robustness against adversarially generated inputs.

To test this hypothesis, we conducted several experiments. First, we ran a number of gradient-based and decision-based adversarial attacks against a baseline model (ResNet32) and tested the efficacy of the resulting adversarial images against the cache models (a similar procedure was recently used in [12]). Specifically, we considered the following four adversarial attacks.

**Fast gradient sign method (FGSM):** This is a standard gradient-based attack [2] where, starting from a test image, we take a single step in direction of the componentwise sign of the gradient vector scaled by a step size parameter $\epsilon$. The step size $\epsilon$ is gradually increased from 0 to a maximum value of 0.5 until the model's prediction of the class label of the perturbed image changes. The image is discarded if no $\epsilon$ yields an adversarial image.

**Iterative FGSM (I-FGSM):** This attack is similar to FGSM, but instead of taking a single gradient step, we take 10 steps for each $\epsilon$ value [9].

**Single pixel attack (SP):** In this attack, a single pixel of the image is set to the maximum or minimum pixel value across the image [15]. All pixels are tried one by one until the perturbed image is classified differently than the original image by the model. If no such pixel is found, the image is discarded.

**Gaussian blur attack (GB):** The image is blurred with a Gaussian filter with standard deviation $\epsilon$ that is increased gradually from 0 to a maximum value of $\max(w, h)$, where $w$ and $h$ are the dimensions of the image in pixels, until the blurred image is classified differently by the model.

We applied each attack to 250 randomly selected test images from CIFAR-10. The attacks were all implemented with Foolbox [14]. For each image, we measured the relative size of the minimum perturbation required to generate an adversarial image as follows [10]:

$$\rho_{\mathrm{adv}}(\mathbf{x}) = \frac{||\mathbf{x}_{\mathrm{adv}} - \mathbf{x}||}{||\mathbf{x}||} \tag{4}$$

where $\mathbf{x}$ denotes the original test image, $\mathbf{x}_{\mathrm{adv}}$ is the adversarial image generated from $\mathbf{x}$ and the norm represents the Euclidean norm. Table 2 reports the mean $\rho_{\mathrm{adv}}$ values for the four

|  | FGSM | I-FGSM | SP | GB |
|---|---|---|---|---|
| $\langle \rho_{\mathrm{adv}} \rangle$ | 0.064 | 0.014 | 0.044 | 0.224 |
| ResNet32 ($\lambda = 0$) | 5.2 | 5.2 | 9.8 | 2.8 |
| ResNet32-Cache3 | **72.8** | 68.4 | 58.8 | 56.0 |
| ResNet32-Cache3-CacheOnly ($\lambda = 1$) | 72.0 | **83.2** | **68.6** | **61.3** |

Table 2: Classification accuracies of different models on adversarial images generated from the CIFAR-10 test set by four different attack methods applied to the baseline ResNet32 model.

|  | FGSM | I-FGSM | SP | GB |
|---|---|---|---|---|
| ResNet32 ($\lambda = 0$) | 0.064 | 0.014 | 0.044 | 0.224 |
| ResNet32-Cache3 | **0.083** | **0.020** | **0.149** | **0.299** |
| ResNet32-Cache3-CacheOnly ($\lambda = 1$) | – | – | – | – |

Table 3: Mean minimum perturbation sizes, $\langle \rho_{\mathrm{adv}} \rangle$, needed for generating adversarial images from the CIFAR-10 test set using direct white-box attacks on the baseline and the cache models. For the cache-only model, we were not able to generate *any* adversarial images using any of the attacks. This is indicated with a '–' in the table above.

attacks. A smaller $\langle \rho_{\mathrm{adv}} \rangle$ value indicates that a smaller perturbation is sufficient to fool the baseline model.

The low classification accuracies for the baseline model (ResNet32) in Table 2 demonstrate that the attacks are indeed effective at generating adversarial images that fool the baseline model. However, different attack methods require different minimum perturbation sizes, as indicated by the varying $\langle \rho_{\mathrm{adv}} \rangle$ values. When the same baseline model is combined with a cache component (ResNet32-Cache3), the classification accuracy on the adversarial images increases significantly. Remarkably, a cache-only model (ResNet32-Cache3-CacheOnly) based on the same baseline model performs even better. This is presumably because the cache-only model alters the decision boundaries of the underlying baseline model more drastically than the linear-combination model. These results show that adversarial images generated for the baseline model do not transfer over to the cache models.

Secondly, we also ran direct white-box attacks on the cache models. Table 3 compares the mean minimum perturbation sizes, $\langle \rho_{\mathrm{adv}} \rangle$, needed for generating adversarial images from the CIFAR-10 test set in different models. In general, generating adversarial images for the cache models proved to be much more difficult than generating adversarial images for the baseline model. For the cache-only model (ResNet32-Cache3-CacheOnly), remarkably, *we were not able to generate any adversarial images using direct white-box attacks*. For the linear-combination cache model (ResNet32-Cache3), we were able to generate adversarial images, but the resulting adversarial images had larger minimum perturbation sizes than the adversarial images generated from the baseline model (Table 3). These results demonstrate the enhanced robustness of the cache models against white-box adversarial attacks.

Adversarial examples often transfer between models [2]: an example created for one model often fools another model as well. To determine whether adding a cache component would impede the transferability of adversarial examples, we presented the adversarial examples generated for the baseline ResNet32 model above to ResNet20 models with or without a cache component. The results are presented in Table 4 and show that adding a cache component to the transfer model significantly reduces the transferability of the adversarial examples (cf. ResNet20 vs. ResNet20-Cache3).

**Cache models behave more regularly near test points in the input space**

As mentioned above, we hypothesized that adding a cache component to a model has a regularizing effect on the model's behavior, because the cache component extends the range in the input space over which the model behaves similarly to the way it behaves near training points and previous work has shown that neural networks behave more regularly near training

|                                                    | FGSM  | I-FGSM | SP    | GB    |
| -------------------------------------------------- | ----- | ------ | ----- | ----- |
| Transfer: ResNet20 ($\lambda = 0$)                 | 77.6  | 89.2   | 70.6  | 54.8  |
| Transfer: ResNet20-Cache3                          | **80.8** | **91.6** | **80.4** | **59.7** |
| Transfer: ResNet20-Cache3-CacheOnly ($\lambda = 1$) | 75.6  | 88.0   | 73.5  | 55.6  |

Table 4: Classification accuracies of ResNet20 variants with or without a cache component on the adversarial examples generated for the ResNet32 baseline model. Adding a cache component reduces the transferability of the adversarial examples.

points than elsewhere in the input space [11]. The results demonstrating the improved robustness of the cache models against various types of adversarial attacks provide evidence for this hypothesis. To test this idea more directly, we calculated the input-output Jacobian, $J(\mathbf{x}) \equiv \partial p(\mathbf{y}|\mathbf{x})/\partial \mathbf{x}$, of different models with or without a cache component at all test points of the CIFAR-10 dataset.

We found that, as predicted, both cache models lead to an overall decrease in the Jacobian norm, $||J(\mathbf{x})||$, averaged over all test points (Figure 4a, inset). The Jacobian norm at a given point can be considered as an estimate of the average sensitivity of the model to perturbations around that point and it was previously shown to be correlated with the generalization gap in neural networks, with smaller norms indicating smaller generalization gaps, hence better generalization performance [11].

Figure 4a shows the mean singular values of the Jacobian for different models averaged over all test points in CIFAR-10. Compared to the baseline model, the linear-combination cache model (ResNet32-Cache3; green) significantly reduces the first singular value but slightly increases the lower-order singular values. On the other hand, although the cache-only model does not reduce the first singular value as much as the linear-combination cache model does, it produces a more consistent reduction in the singular values (ResNet32-CacheOnly; black). Figure 4b-c shows this differential behavior of the two cache models by plotting the Jacobian norms at individual test points. In Figure 4b, the points are clustered below the diagonal for high $||J(\mathbf{x})||$ values and above the diagonal for low $||J(\mathbf{x})||$ values, indicating that the linear-combination cache model reduces the Jacobian norm in the first case, but increases it in the second case. In Figure 4c, on the other hand, the points are more consistently below the diagonal, suggesting that the cache-only model reduces the Jacobian norm in both cases. This pattern is consistent with the singular value profiles shown in Figure 4a.

We conjecture that the different singular value profiles of the two cache models may be related to their different generalization patterns. We have observed above that the linear-combination cache model has a better test accuracy than the baseline and cache-only models (Table 1), but the cache-only model is more robust to adversarial perturbations (Table 2) than the linear-combination cache model. If one considers the test accuracy as measuring the within-sample or on-the-data-manifold generalization performance and the adversarial accuracy as measuring the out-of-sample or off-the-data-manifold generalization performance of a model, the small first singular value may explain the superior test accuracy of the linear-combination cache model, whereas the small lower order singular values may explain the superior adversarial accuracy of the cache-only model. We leave a fuller exploration of this hypothesis to future work.

## 3 Discussion

In this paper, we proposed a simple method to improve the classification performance of large-scale image recognition models at test time. Our method relies on the observation that higher layers of deep neural networks contain independent class-relevant information that is not already contained in the output layer of the network and this extra information is in an easily extractable format. In this work, we have used a simple continuous cache-based key-value memory to extract this information. This particular method has the advantage that it does not require any re-training or fine-tuning of the model. Moreover, we showed that it also significantly improves the robustness of the underlying model against adversarial attacks.

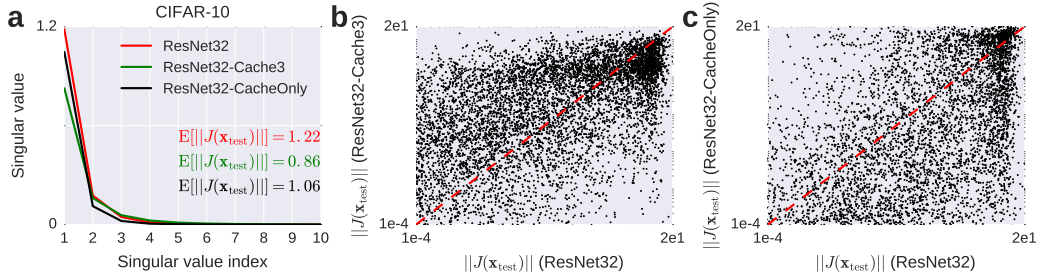

Figure 4: Cache models behave more regularly near test points in the input space. (a) Mean singular values of the Jacobian at test points for three different models. The mean Jacobian norm (averaged over test points) for each model is indicated in the inset. (b) and (c) show the Jacobian norms at individual test points.

However, there may be alternative ways of extracting this extra class-relevant information. For example, linear read-outs can be added on top of the layers containing this extra information and then trained on the training data either separately or in conjunction with read-outs from other layers, while the rest of the network remains fixed.

We used deep ResNet and DenseNet architectures in our simulations. These architectures were specifically designed to make it easy to pass information unobstructed between layers [5–7]. It is thus noteworthy that we were able to extract a significant amount of extra class-relevant information from the non-final layers of these networks with a relatively simple read-out method. We expect the gains from our method to be even larger in deep networks that are not specifically designed to ease the information flow between layers, e.g. networks without skip connections.

### Related work

Our cache model was directly inspired by the work of Grave et al. [3], where a conceptually very similar model was first introduced in the context of language modeling. In language modeling, prediction of rare words that do not occur frequently in the training corpus poses a problem. Grave et al. [3] proposed extending the standard recurrent sequence-to-sequence models with a continuous key-value cache that stored the recent history of the recurrent activations in the network as keys and the next words in the sequence as values. The primary motivation in this work was to address the rare words problem using the basic idea that if a word that is overall rare in the corpus appeared recently, it is likely to appear again in the near future. Hence, by storing the recent history of activations and the corresponding target words, one can quickly retrieve the correct label from the recent context when the rare word reappears again.

A similar motivation might apply in image recognition tasks as well. Although there is no temporal context in these tasks, a similar "rare features" problem arises in image recognition too: if correct classification of an item depends on the detection of a set of distinctive features that do not occur very frequently in the training data, standard image recognition models trained end-to-end with gradient descent might have a difficulty, since these models would typically require a large enough number of examples to learn the association between those features and the correct label. By storing those features and the corresponding labels in a cache instead, we can quickly retrieve the correct label upon detection of the corresponding features and hence circumvent the sample inefficiency of end-to-end training with gradient descent.

Similar cache models have recently been proposed to improve the sample efficiency in reinforcement learning [13] and one-shot learning problems [8] as well. In these cases, however, the cache component (i.e. its key-value pairs) was trained jointly with the rest of the model, hence these models are different from our model and the model of [3] in this respect.

Two recent papers have used cache memories to improve the robustness of image recognition models against adversarial attacks [12, 18]. In [18], inputs are projected onto the data manifold approximated by the convex hull of a set of features. These features, in turn, are formed from a set of "candidate" items retrieved from a cache. Unlike our model, however, both the features and the projection operator are trained jointly with the rest of the model and the model is further constrained to behave linearly on the convex hull through mixup training [17]. Overall, this model is significantly more complicated than the simple cache model proposed in this paper. However, the basic mechanism behind its improvement of robustness against adversarial examples is similar to ours.

Our model is more similar to the deep $k$-nearest neighbor ($k$-NN) model introduced in [12]. In this model, for a given test item, $k$ nearest neighbors from the training data are retrieved based on their representations at each layer of a deep network. The model's prediction for the test item as well as a confidence score are then computed based on the retrieved nearest neighbors and their labels. There are two main differences between our model and the deep $k$-NN model. First, the deep $k$-NN model retrieves $k$ nearest neighbors (typically, $k = 75$ nearest neighbors were used in the paper) and weighs them equally in the prediction and confidence computations, whereas our model uses a continuous cache that utilizes all the items in the cache and weighs them by their similarity to the representation of the test item. This is an important difference for the adversarial robustness of these two models, since $k$-NN models with a large $k$ are known to be more robust against adversarial examples than $k$-NN models with a small $k$ [16]. Secondly, the deep $k$-NN model uses representations at all layers in retrieving the nearest neighbors, whereas our model uses only a small number of layers close to the output of the network. We have presented evidence suggesting that using the earlier layers might adversely affect the generalization performance of the model by making it vulnerable to surface similarities that are not relevant for the classification task (Figure 2).

**Future directions**

For the sake of simplicity, we have used the entire layer activations as key vectors in our cache model. Since these vectors are likely to be redundant, a more efficient alternative would be to apply a dimensionality reduction method first before storing these vectors as keys in the cache component. The simplest such method would be using random projections [1], which has favorable theoretical properties and is easy to implement. This would allow us to test larger cache sizes and more layers in the key vectors. Using large cache sizes is important especially in problems with large training set sizes, such as ImageNet, as we empirically observed this to be a more important factor affecting the generalization accuracy than the number of layers used in the key vectors. Relatedly, efficient nearest neighbor methods can be utilized to scale our model to effectively unbounded cache sizes which would be useful under online learning and/or evaluation scenarios [4].

When more than one layer was used in the cache component, we selected the layers based on manual exploration. More principled ways of searching for combinations of layers to be used in the cache component should also improve the generalization performance of the cache models.

We have only considered classification tasks in this paper, but a continuous key-value cache component can be added to models performing other image-based tasks, such as object detection or segmentation, as well. However, different tasks might require different similarity measures (Equation 1) and different ways of combining the predictions of the cache component with the predictions of the underlying model (Equation 3). Video-based tasks are also obvious candidates for the application of a continuous cache component as the original paper [3] that motivated our work also used it in a temporal task, i.e. sequence-to-sequence modeling.

# Acknowledgments

I thank the staff at the High Performance Computing cluster at NYU, especially Shenglong Wang, for their excellent maintenance efforts and for their help with troubleshooting.

## Footnotes

[1]Table 1 reports the error rates to make comparison with previous results on these benchmark datasets easier. In the remainder of the paper, we will report classification accuracies instead.

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
