[Reviews · NeurIPS 2018]

Reviewer 1



This paper presents a cache model to be used in image recognition tasks. The authors argue that class specific information can be retrieved from earlier layers of the network to improve the accuracy of an already trained model, without having to re-train of finetune. This is achieved by extracting and caching the activations of some layers along with the class at training time. At test time a similarity measure is used to calculate how far/close the input is compared to information stored in memory. Experiments show that performance is improved in CIFAR 10/100 and ImageNet. In addition the authors show that a cache model increases resistance in adversarial attacks and show that experimentally that the cache model is more stable at larger regions around the training data. The paper is well written and contains a thorough experimental design. I would like to see a more conventional organisation of the paper. It feels off to not read previous research and jump straight at the results. In general I liked reading the paper. The authors hypothesize that using a cache model similar to work of Grave et al. can benefit image recognition tasks as well. The motivation is clearly explained and the contributions are easy to extract. I liked the idea that using a cache model is like putting a prior on data so that the model prefers inputs similar to the training set. How would that work in a real world case where data is not that perfect though as in benchmark datasets? I feel that favouring similar data to training set might not be usable in datasets in the wild where some of the assumptions on data break. I do have some concerns over computation. Very deep networks will require searching in a combinatorial fashion for number of layers, which layers, what should lambda and theta be set to etc. I think a discussion on this should be present and ways to go around it. Perhaps this can be something that is learnt concurrently to the training procedure. Table 2 shows p_adv. This was tuned to the baseline model and tested at cache3 and cache-only using that setting. I would like to see what happens when p_adv is tuned on the other models too. The main issue I have is that by adding a cache model, essentially you have changed the model and how decisions are made. So resnet32-cache3 is a different model than vanilla resnet32. Looking at how adversarial attacks transfer with resnet20 for instance, though there is an impact it's not nearly as catastrophic as in resnet32. So I am left feeling that maybe the performance of cache models is because it is more like transfer and can't judge with certainty if the adversarial behaviour is indeed there. Perhaps tuning p_adv for cache3 and cache-only will provide more information on how large a perturbation needs to be to break the model. Regarding references, there's a lot of arxiv work when most have been published to peer reviewed journals and conference. There should be a second pass on that to fix it. * The rebuttal has addressed most of my concerns and cleared up a few other places.

Reviewer 2



This paper studies how to improve the test performance of an already trained model for image recognition, without retraining or fine-tuning. The motivation is based on an empirical observation that the layers close to the output may contain independent, easily extractable class-relevant information of target images. Thus, this paper extends a simple cache model, first published in the reference [3], to the image recognition tasks, and discusses several insights about the proposed image recognition model. The empirical studies on three classic datasets demonstrate the effectiveness and robustness of the proposed method. Positive points: 1. The proposed method succeeds to exploit the information from the layers close to the output, through a simple cache technique, to improve the performance and robustness of image recognition models. 2. The analyses about the adversarial experiment and regularizing effect are insightful. Negative points: 1. The originality of the proposed method, to some extent, is limited. 2. The empirical observation about the superiority of high-level features is not new, since several existing studies have found it and took advantage of it. [1] Photo-Realistic Single Image Super-Resolution Using a Generative Adversarial Network [2] Perceptual Losses for Real-Time Style Transfer and Super-Resolution 3. It hard to choose the number of layers to concatenate the activations. 4. The analysis, i.e., the discussion of superficial similarities, about why the layers close to the output contain class-relevant information, is insufficient and lacks depth. It would be better to analyze this empirical finding deeper. 5. Although the adversarial experiment, to some extent, shows some generalization performance, the persuasion is insufficient. In my view, the success of the cache technique mainly attributes to the transferability of the intrinsic discriminative spaces between training and testing data. Hence, whether this method can generalize to the case, where the training and testing data come from different datasets, is important and remains unknown. Suggestions: 1. The scheme of the cache model is not clear enough. It is better to improve it. 2. Typos: in the bracket of line 199, (ResNet32-Cache3; black) seems to be (ResNet32-CacheOnly; black).

Reviewer 3



The authors propose a cache memory model for image recognition, extending the one proposed by Grave et al. for language modeling. This simple extension to standard CNNs for image classification provides a boost in accuracy and improves the robustness to adversarial attacks. An analysis for these results is provided, based on the input-output Jacobian. The method is powerful, conceptually simple and easy to implement. In my opinion, it may open new interesting research lines, as this work can be extended to other domains, e.g. few-shot or lifelong learning. I enjoyed reading the manuscript, it is easy to follow and the design choices are clearly motivated. Overall, I consider this submission a very positive contribution to NIPS. Please find below some minor modifications that I would like to see in the revised version of the manuscript: - Did you run any experiment on CIFAR-10/100 to check the effect of reducing the number of items stored in the cache? Memory footprint is one of the main drawbacks of this method, which can prevent its deployment for larger scale datasets. Understanding the role of the cache size is thus important, and some plot/table with this information would add value to the paper. - Related to the previous point, did you perform any experiment where images in the training set are added (or not) to the cache only if they are classified correctly? For memory-constrained applications, it would be important to properly select which samples are kept in the cache. - Regarding the adversarial attacks in Table 2, I understand that they are generated for the baseline model and then evaluated on the same model combined with a cache model. This is somehow measuring transferability from the baseline model to the cache-enhanced one. Wouldn’t it be possible to generate the adversarial images directly for the cache-enhanced models? - Line 175 points to Table 2, but I believe it should link to Table 3. - Seeing that Section 2 is already ‘Results’ was a bit surprising. Perhaps the authors could split it into ‘Method’ and ‘Results’ and follow a more traditional structure. ------------------------------------------------------------------------------------------------------------------------------------------------------------------------------------------ The authors addressed (or will adress) all my concerns/questions to improve their submission. As stated in my review, I vote for accepting this submission.